# Betaine Alleviates Bisphosphonate-Related Osteonecrosis of the Jaw by Rescuing BMSCs Function in an m6A-METTL3-Dependent Manner

**DOI:** 10.3390/ijms26115233

**Published:** 2025-05-29

**Authors:** Yizhou Jin, Jiaxin Song, Zhanqiu Diao, Xiao Han, Zhipeng Fan

**Affiliations:** 1Beijing Key Laboratory of Tooth Regeneration and Function Reconstruction, Beijing Stomatological Hospital, School of Stomatology, Capital Medical University, Beijing 100050, China; jinyizhou123@126.com (Y.J.); songjiaxin2025@163.com (J.S.); dzq990922@163.com (Z.D.); hxiao0418@163.com (X.H.); 2Beijing Laboratory of Oral Health, Capital Medical University, Beijing 100069, China; 3Research Unit of Tooth Development and Regeneration, Chinese Academy of Medical Sciences, Beijing 100730, China

**Keywords:** betaine, bisphosphonate, N6-adenosine methylation, mesenchymal stem cells, zoledronic acid

## Abstract

Bisphosphonate-related osteonecrosis of the jaw (BRONJ) is one of the side effects of bisphosphonate (BP) administration. Despite some preventive measures having been suggested, a definitive and effective treatment strategy for BRONJ remains to be established. Recent evidence has indicated that BPs dramatically impair the function of orofacial bone marrow stromal cells (BMSCs), which may contribute to the development of osteonecrosis. Thus, we hypothesized that recovery-impaired function of BMSCs at lesion sites could be beneficial in treating BRONJ. N6-methyladenosine (m6A) modification is the most common epigenetic modification and has been demonstrated to play a vital role in the modulation of BMSCs’ function. We detected the role of m6A modification in regulating the function of orofacial BMSCs under BP stimulation, and found that BPs led to a reduction in the global m6A methylation level, SAM level, and METTL3 expression in BMSCs during the osteogenic differentiation period. Meanwhile, betaine, a methyl group donor, effectively reversed the BP-decreased global m6A methylation level and SAM level in BMSCs, as well as rescuing the differentiation ability of impaired BMSCs. In the last part, we built a BRONJ rat model and supplemented rats with betaine via drinking water. The results showed that betaine successfully attenuated bone lesions and promoted wound healing in BP-injected rats, thereby providing new insight into future clinical treatment for BRONJ.

## 1. Introduction

Bisphosphonates (BPs) are a class of chemically synthesized drug that are widely used in various orthopedic diseases such as osteoporosis and in the prevention of cancer bone metastases [1]. There are, however, unwanted side effects. One of these is bisphosphonate-related osteonecrosis of the jaw (BRONJ), which is defined as exposed mandible bone in the oral environment for more than 8 weeks [2]. The exact cause of BRONJ is not well understood, but it usually occurs when invasive procedures, such as tooth extraction, damage the alveolar bone [3]. Although some preventive measures, such as oral care before tooth extraction, have been suggested, a clear and effective treatment strategy for BRONJ has not been established [4].

The process of osteogenesis is important for bone lesion repair and bone marrow-derived mesenchymal stem cells (BMSCs) are significant osteoblast progenitors. It has been reported that BMSCs derived from the peripheral and central areas of BRONJ lesions exhibit poor multi-differentiation ability, contributing to the suppression of bone lesion repair in BRONJ patients [5]. Therefore, the activity of recovery BMSCs at the lesion area may become useful in treating BRONJ. Recently, cell therapy using BMSCs has been demonstrated to be effective in treating BRONJ [6,7]; however, these exogenous BMSCs usually do not survive long at the lesion site and their clinical efficacy is influenced by various factors, such as the age of the donor [8,9,10]. Thus, restoring the native BMSC function at the lesion area of BRONJ may have a more effective outcome than exogenous BMSC transplantation.

N6-methyladenosine (m6A) modification is the most common epigenetic modification [11], and is regulated by methyltransferases, demethylases, and methyl-binding proteins [12]. Accumulating evidence has demonstrated that m6A modification plays a vital role in the modulation of the function of BMSCs and in the progression of various orthopedic disorders, including osteoporosis, osteoarthritis, and osteosarcoma [13,14,15]. For example, METTL3, one of methyltransferases, promoted the survival of BMSCs in the steroid-induce osteonecrosis of the femoral head (SONFH) lesion area [16]. Our previous in vitro study showed that METTL7A, another methyltransferase, rescued BP-induced BMSCs from dysfunction via regulating the m6A modification of the osteogenesis-related gene, providing evidence for the role of m6A modification in BRONJ progression [17].

Betaine (N,N,N-trimethylglycine), known to function physiologically as an important methyl group donor, is widely distributed in animals, plants, and microorganisms [18]. The transmethylation reaction of betaine is critical for S-adenosyl methionine (SAM) synthesis, and SAM regulates m6A modification and influences gene expression [19,20]. Studies from another group have revealed that betaine promoted the osteogenic differentiation of hAD-MSCs and could alleviate alcohol-induced osteonecrosis of the femoral head, but the mechanism remains unclear [21,22]. Moreover, whether betaine has an osteoprotective effect on BMSCs under BPs administration remains unclarified.

In this study, we hypothesized that betaine could alleviate BP-induced osteonecrosis by rescuing the function of BMSCs in an m6A-dependent manner. To test this hypothesis, BPs were administered to orofacial BMSCs 3 days before the induction of osteogenic differentiation, followed by betaine application. Also, an in vivo study was performed, in which a rat BRONJ model was created and betaine was supplemented to the rat model via drinking water. We studied the osteoprotective effect of betaine and its modulation of RNA m6A methylation in BMSCs by conducting histological and molecular biological evaluation.

## 2. Results

### 2.1. BPs Impaired BMSCs Function and Decreased Global m6A Methylation Level of BMSCs

BPs are divided into nitrogen and non-nitrogen BPs. Zoledronate acid (ZA) is a nitrogen-containing third-generation BP which is most widely used in clinical treatment [1]. We applied ZA in this study and detected its impact on BMSCs’ osteogenic differentiation. Orofacial BMSCs were extracted from rats’ mandible bones following the rats receiving an intraperitoneal injection of ZA or saline (0.1 mg/kg) for 6 weeks (3 times/week). Compared to the BMSCs extracted from saline-injected rats (Control Group), the ALP/ARS staining and calcium ion quantification showed weaker signals in the BMSCs extracted from ZA-injected rats (BPs Group) (Figure 1a,b). Consistent with the ALP/ARS staining, the protein expression of OSX/OCN in the BPs Group was decreased as well (Figure 1c). Then, we investigated the impact of ZA in vitro. Human orofacial BMSCs were isolated from donors’ mandible bones and pretreated with or without ZA for 3 days before the induction of osteogenic differentiation. We found that the ZA dramatically suppressed the osteogenic differentiation potential of BMSCs in vitro, as evidenced by weaker signals obtained in ALP/ARS staining and calcium ion quantification, as well as a reduced OSX/OCN expression level in the ZA Group (Figure 1d–f). In addition, ZA-pretreated BMSCs were mixed with HA/TCP materials and transplanted subcutaneously into the backs of mice. Compared to the CTR Group, the HE/Masson staining images indicated less generation of bone-like tissues in the ZA Group (Figure 1g–i). Similarly, immunohistochemical staining and quantitative measurements showed less OCN-positive bone matrix in the ZA Group (Figure 1j,k). The above results suggested the negative impact of BPs on BMSCs’ osteogenic differentiation ability.

Accumulating evidence has demonstrated that m6A modification plays a vital role in the modulation of BMSCs’ function and the progression of various orthopedic disorders [13,14,15]. With regard to how BPs negatively influenced BMSCs’ differentiation ability, we hypothesized that m6A modification was involved. To test this hypothesis, we verified the m6A methylation level in BP-treated BMSCs. BMSCs were extracted from saline or ZA-injected rats and cultured in osteogenic induction medium. At day 3 of osteogenic differentiation, the confocal images labeled with anti-m6A antibody in the BPs Group had lower values than those in the Control Group (Figure 2a,b). Similarly, by conducting a colorimetric m6A quantification assay, we observed that the global m6A methylation level in the BPs Group was lower than that in the Control Group at day 7 of osteogenic differentiation (Figure 2c). We then detected the impact of ZA on the m6A methylation level in vitro as well. Human orofacial BMSCs were pretreated with or without ZA for 3 days, and were then cultured with growth medium (GM) or osteogenic induction medium (OM). At day 3, the confocal images labeled with anti-m6A antibody had higher values in the CTR-OM Group compared to those in the CTR-GM Group, whereas decreased in ZA-OM Group compared to the CTR-OM Group (Figure 2d,e). At day 7, the detection of the global m6A methylation level in BMSCs showed the same trends as the confocal images (Figure 2f). These results indicated that the BPs significantly decreased the m6A methylation level of the BMSCs during osteogenic differentiation process.

### 2.2. Betaine Rescued BPs-Induced BMSCs Dysfunction In Vitro

Betaine contains three chemically reactive methyl groups, and its methyl group is transferred by betaine-homocysteine methyltransferase (BHMT) to form methionine. Methionine is subsequently converted into SAM, which plays a crucial role in the methionine cycle and usually serves as an indicator for transmethylation potential [23]. Considering the lower m6A methylation level that was observed in the ZA-treated BMSCs, we applied betaine to validate whether methyl group donors could rescue deficient m6A methylation and reverse the inhibitory effect of BPs on BMSC osteogenesis. We pretreated human orofacial BMSCs with or without ZA for 3 days and then cultured the cells in osteogenic induction medium with or without 10 mM betaine. A colorimetric quantification assay was performed on day 7 to detect the global m6A methylation and SAM level of the BMSCs. The quantification results showed that betaine reversed the ZA-induced decrease in the m6A methylation and SAM level of the BMSCs (Figure 3a,b). Similarly, confocal images labelled with m6A-antibody on day 3 showed more m6A-positive BMSCs in the ZA+Betaine Group (Figure 3c,d). Next, we explored the role of betaine in regulating BMSCs’ differentiation ability. Compared to the CTR Group, the ALP/ARS staining and calcium ion quantification in the Betaine Group showed stronger signals, indicating that betaine promoted orofacial BMSCs’ osteogenic differentiation ability (Appendix A). Then, we designed a rescue experiment. Compared to the ZA Group, the ALP/ARS staining and calcium ion quantification showed stronger signals in the ZA+Betaine Group, as well as a higher expression level of OSX/OCN (Figure 3e–g). In addition, the in vivo bone regeneration results showed more bone-like tissues in the ZA+Betaine Group than in the ZA Group (Figure 3h,i). Similarly, immunohistochemical staining images showed more OCN-positive bone matrix in the ZA+Betaine Group (Figure 3h,j). The above results indicated that betaine rescued the BP-suppressed m6A methylation level and the differentiation ability of orofacial BMSCs.

### 2.3. Betaine Alleviated Osteonecrosis of the Jaw in BRONJ Rat Model

Given the osteoprotective effect of betaine on BP-treated orofacial BMSCs, we speculated that betaine might attenuate BP-related osteonecrosis of the jaw. To test this hypothesis, we established a BRONJ rat model and supplemented the rat with betaine via drinking water. The establishment of the BRONJ rat model was confirmed by the fact that the extraction socket was not covered with the oral mucosa and the alveolar bone was exposed at 8 weeks post-tooth extraction [10].

We first explored whether betaine alleviated the impairment at the early stage. At 2 weeks post-tooth extraction, we extracted BMSCs from the rats’ mandible bones and then conducted a colorimetric quantification assay. The results revealed that the global m6A methylation and SAM level in the BPs+Betaine Group were higher than in the BPs Group, indicating that betaine reversed the BP-decreased m6A methylation in vivo (Figure 4b,c). Meanwhile, there was no covering of the epithelium on the extraction socket or exposed alveolar bone in the BPs Group rats, while socket was almost covered with soft tissue and there was no obvious bone exposure in the BPs+Betaine rats (Figure 4d,e). Moreover, the slides of mandible bone were HE- and Masson-stained. Compared to the BPs Group, the representative images of the BPs+Betaine Group showed a decreased empty lacunae area and less necrotic bone (%), with a simultaneously increased blue-stained trabecular area in the sockets (Figure 4f–j). The above results demonstrated that betaine alleviated BP-induced impairment at the early stage.

We next explored whether betaine alleviated the impairment at 8 weeks post-tooth extraction. Similarly, we extracted BMSCs from rats’ mandible bones and detected the global m6A methylation and SAM level. The quantification results indicated that betaine reversed the m6A methylation and SAM level in the BMSCs at 8 weeks post-extraction (Figure 5a,b). At 8 weeks post-extraction, the BPs Group rats still showed open and inflamed wounds, while the BPs+Betaine Group rats presented complete mucosal coverage without inflammation or swelling (Figure 5c,d). When comparing each sample by their μCT, we found that there was no bone-like tissue formed in the extraction sockets of the BPs Group rats, whereas more hard tissue was generated in the BPs+Betaine Group (Figure 5e–g). In addition, histological staining was conducted again to evaluate the healing of the osseous tissue. Compared to the BPs Group, the H&E/Masson staining confirmed the continuity of the epithelium, that more hard tissue was formed, and that there was less necrotic bone (%) in the BPs+Betaine Group at 8 weeks (Figure 5h–k). These results demonstrated that betaine alleviated BP-related osteonecrosis at 8 weeks post-tooth extraction.

### 2.4. Betaine Regulated BMSCs Function in an m6A-METTL3-Dependent Manner

It was observed that betaine functioned physiologically as a methyl group donor, and, in the last part of our study, we further explored how methyl group donors regulated BMSCs’ function. m6A is catalyzed by the methyltransferase complex and METTL3 is the core methyltransferase of the m6A writer complex [13,24,25]. We detected the expression level of METTL3 in human orofacial BMSCs and found that BPs decreased the METTL3 expression, while betaine reversed the decrease (Figure 6a). We then constructed shRNA-mediated METTL3 knockdown BMSCs (sh-METTL3) (Figure 6b,c). In sh-NC BMSCs, the ZA+Betaine Group showed stronger signals of ALP/ARS staining and calcium ion quantification, as well as a higher expression level of OSX/OCN, than the ZA Group. However, in sh-METTL3 BMSCs, the ZA+Betaine Group showed similar signals of ALP/ARS staining and calcium ion quantification, as well as a similar expression level of OSX/OCN, to the ZA Group, indicating that betaine failed to rescue the BP-induced impairment in the sh-METTL3 BMSCs (Figure 6d–f). Taken together, these results demonstrated that betaine rescued BMSCs’ function in an m6A-METTL3-dependent manner (Figure 6g).

## 3. Discussion

BMSCs are significant osteoblast progenitors and their osteogenesis function is important for bone lesion repair. Attempts at BMSCs transplantation have been reported to be successful in treating BRONJ animal models, but in these studies the physical function of BMSCs was attributed to immune modulation and the changes in their osteogenic differentiation ability have been ignored for many years [6]. In 2017, He et al. demonstrated that native BMSCs did exist at a BRONJ lesion area and that their self-renewal and multi-differentiation ability were dramatically impaired [5]. In our previous in vitro study, we found that the impact of BPs on BMSCs seemed double-edged, as 0.1 μM of BPs slightly promoted their proliferation and differentiation ability, while 5 and 10 μM of BPs inhibited their activities [17]. Some researchers estimated that the average concentration of BPs at patients’ bone lesion sites was as high as 1 mM, which is much higher than the concentration of BPs needed to affect BMSCs’ morphology and suppress their activities in vitro [26,27]. In this study, we found that BMSCs extracted from the mandible bones of BP-injected rats showed impaired osteogenic potential compared to Control BMSCs, which is consistent with He’s finding. Moreover, we pretreated human orofacial BMSCs with 5μM BPs and found that their activity was significantly inhibited. These in vitro results were consistent with our previous findings, and we chose the concentration of 5μM in the following study.

Due to the impaired osteogenic activities that were observed, we hypothesized that recovery of the BMSCs’ dysfunction at the BRONJ lesion area might become a potent treatment strategy for BRONJ. Based on the background that numerous reports have shown a close relationship between m6A modification and BMSCs’ function, we supposed that m6A modification plays a vital role in regulating orofacial BMSCs at the BRONJ lesion area. To identify this hypothesis, we conducted a colorimetric m6A quantification assay and m6A immunofluorescence staining. The results showed a decreased global m6A methylation level in BMSCs extracted from BP-injected rats, and human BMSCs under in-vitro BP administration also witnessed decreased m6A methylation levels during the osteogenic differentiation process. It is necessary to mention that the impact of BPs on the m6A methylation level seems to differ in different cell lines. Yang et al. pointed out that the global m6A methylation level and the expression of METTL14 (one of the m6A methyltransferases) were strongly increased in RAW264.7 (one of the osteoclast precursor cells) under BP stimulation [28], which is opposite to our findings in orofacial BMSCs.

Considering the low m6A methylation level observed in BP-treated BMSCs, we applied a methyl group donor, betaine, in the following study. We demonstrated that betaine regulated orofacial BMSCs’ activities in an m6A-dependent manner, which is consistent with Yang’s finding in HMC3 cells [29]. Under betaine supplementation, the BP-induced reduction in global m6A methylation and the SAM level were reversed, as was the impaired BMSCs differentiation ability. Based on the in vitro osteoprotective effect, we supplemented betaine in a BRONJ rat model via drinking water to explore its in vivo efficacy. At 2 weeks post-tooth extraction, BRONJ rats supplemented with betaine showed fewer empty lacunae and less necrotic bone, indicting that betaine prevented the progression of BRONJ at the early stage of tooth extraction. We also noticed that the soft tissue healing was accelerated, but the mechanism underlying betaine promoted fibroblast activity remained to be further explored. At 8 weeks post-extraction, the extraction sockets of betaine-supplemented rats were almost fulfilled with newly formed hard tissue and covered with epithelium without bone exposure, indicating that betaine successfully prevented the occurrence of BRONJ. Considering that betaine could be easily administered via drinking water, it would be more helpful for future clinical trials to compare this substance to other reported mitigants, such as beta-tricalcium phosphate and extracellular vesicles [3,10]. The osteoprotective effect of betaine has been reported in some other studies. For example, an in vitro study demonstrated that betaine promoted the osteogenic differentiation ability of hAD-MSCs [16]. Moreover, some in vivo studies applied betaine and successfully treated orthopedic disorders in animal models, including osteoarthritis, chemotherapy-induced bone loss, and alcohol-induced osteonecrosis [22,30,31]. However, the molecular mechanism underlying how betaine protected and promoted the observed osteogenic differentiation activity was not explored in these studies.

m6A is catalyzed by the methyltransferase complex consisting of an enzymatic subunit METTL3, a substrate recognition subunit METTL14, and a regulatory subunit WTAP [13]. METTL3 is the core methyltransferase of the m6A writer complex and was proven to regulate osteogenic differentiation in our previous study [24,25]. In addition, Wu et al. demonstrated that METTL3 overexpression in MSCs protected mice from estrogen deficiency-induced osteoporosis [13]. To explore the molecular mechanism underlying betaine’s promotion of osteogenic differentiation, we detected the expression of METTL3 and constructed shRNA-mediated METTL3 knockdown BMSCs. The expression of METTL3 was decreased both in BP-treated human BMSCs and BMSCs at the lesion area of BRONJ rats, whereas the decrease was reversed in betaine-supplemented human BMSCs and BMSCs at the lesion area of betaine-supplemented BRONJ rats. Moreover, betaine failed to rescue the impaired differentiation ability of sh-METTL3 BMSCs, further indicating that the betaine regulated BMSCs’ differentiation ability in an m6A-METTL3-dependent way, giving us the hope that METTL3 may be another potent treatment target to alleviate BP-induced impairment. However, our findings demonstrated that betaine significantly enhanced the overall m6A levels in BMSCs, although the downstream effects genes which were betaine-methylated have not been proven in this study, and this requires further investigation. 

In summary, we demonstrated that betaine attenuated BRONJ via rescuing orofacial BMSCs’ differentiation ability. Specifically, orofacial BMSCs extracted from BP-injected rats showed an impaired osteogenic differentiation ability and a lower m6A methylation level. Additionally, an in vitro study indicated that human BP-pretreated BMSC’ witnessed a decreased osteogenic differentiation ability as well as a lower m6A methylation level. However, when betaine, a methyl group donor, was administered, the dysfunction of BMSCs was halted and recovered to the extent of the non-BPs pretreated group. In vivo, betaine supplemented in a BRONJ rat model successfully prevented bone exposure and promoted new bone generation in extraction sockets. Our research advanced the potential nexus between epigenetic regulation and BP-induced impairment. Targeting the m6A methylation in the osteogenic process of orofacial BMSCs might represent a potential therapeutic approach to alleviating bone necrosis.

## 4. Materials and Methods

### 4.1. Cell Culture and Drug Administration

Orofacial BMSCs were isolated from jaw bone as previously described [17]. The tissue collection followed the ethics guidelines of the Ethics Committee of Beijing Stomatology Hospital (Approval number: CMUSH-IRB-KJ-PJ-2023-40). Briefly, the bone tissues were thoroughly rinsed and digested for 30 min. The suspension was collected and seeded in culture medium (ɑ-MEM, Gibco, Grand Island, NY, USA) supplemented with 1% penicillin/streptomycin and 10% FBS. The cells were continuously cultured, and passages 3–5 were used in the following study.

For osteogenic induction, BMSCs were treated with osteogenic medium supplemented with 5 mM β-glycerophosphate, 50 μg/mL ascorbic acid, and 100 nM dexamethasone. Zoledronic acid (SML0223, Sigma-Aldrich, St. Louis, MO, USA) and betaine (B2629, Sigma-Aldrich, St. Louis, MO, USA) were dissolved in sterile water and stored at a concentration of 5 mM and 10 M respectively. A total of 5 μM Zoledronic acid was added to the culture medium 3 days before inducing osteogenic differentiation. A total of 10 mM betaine was added 3 days before inducing osteogenic differentiation and continuously applied during the whole induction process.

### 4.2. Animals and Treatments

All experimental procedures involving animals were conducted following the ethics guidelines approved by the Beijing Stomatology Hospital Animal Ethics Committee (Approval number: KQYY-202307-003).

Thirty-six male Sprague Dawley rats (180–200 g) were obtained from Vital River Experimental Animal Technique Company, China. After 1 week acclimation, rats received intraperitoneal injection of Zoledronic acid or saline (0.1 mg/kg) for 6 weeks (3 times/week). Six weeks later, the first maxillary molar of all rats was extracted under general and local anesthesia. The animals were randomly assigned to the following 3 groups: (1) Control: rats received saline injection as described above; (2) BPs: rats received Zoledronic acid injection; and (3) BPs+Betaine: rats received Zoledronic acid injection and were supplemented with betaine (2% *w*/*v*) via drinking water from the day of tooth extraction. At 2 weeks and 8 week post-tooth extraction, rats were deeply anesthetized and sacrificed. The upper jaw was removed for preparation of histological specimens (paraffin-embedded) and X-ray computed microtomography (µCT) imaging. µCT was taken prior to HE/Masson staining with paraffin-embedded specimens. All experiments were n = 6 per group.

### 4.3. Transplantation in Nude Mice

Twenty-five female NU/NU mice (8 weeks old, Vital River Experimental Animal Technique Company, Beijing, China) were used in this study. 2 × 10^6^ BMSCs were mixed with 20 mg HA/tricalcium phosphate and then transplanted to mice subcutaneously. The samples were harvested at 8 weeks post-transplantation and HE/Masson/immunohistochemical staining was performed as previously described [32]. All experiments were n = 5 per group.

### 4.4. Alkaline Phosphatase and Alizarin Red Staining Detection

BMSCs were cultured in osteogenic medium which was changed every 3 days. ALP nodes were stained by ALP Staining Kit (C3206, Beyotime, Shanghai, China) according to manufacturer’s instructions at 7 days post-osteogenic differentiation. Mineralization nodes were stained by Alizarin Red at 14 days post-osteogenic differentiation. Briefly, cells were fixed with 4% paraformaldehyde for 30 min and stained with 1% Alizarin Red solution for 1–3 min. The mineralization nodes were then dissolved by 10% cetylpyridinium chloride solution, and the calcium quantification was read at a wavelength of 562 nm (OD562).

### 4.5. RNA Extraction and Real-Time PCR

TRIzol reagent was used to extract total RNA from BMSCs. Reverse-transcription Kit (R333, Vazyme, Nanjing, China) was applied to reverse total RNA into cDNA. GAPDH was used to normalize mRNA levels. The primer sequences are listed in Appendix A.

### 4.6. Western Blot

Total proteins were extracted as previously described [32]. The primary antibodies were as follows: anti-OSX antibody (bs25532R, Bioss, Beijing, China), anti-OCN antibody (bsm62874R, Bioss, Beijing, China), anti-METTL3 antibody (15073-1-AP, Proteintech, Wuhan, China), and anti-β-actin antibody (TA-09, ZSGB-BIO, Beijing, China).

### 4.7. m6A-ELISA

RNA m6A methylation level was detected by the m6A Methylation Quantification Kit (Colorimetric) (ab185912, Abcam, Cambridge, UK). Briefly, following addition of 80 μL binding solution to each well, 200 ng sample (1–8 μL) RNA was added, mixed, and incubated at 37 °C for 90 min. The Binding Solution was then removed, following incubation of sample RNA with Capture Antibody at room temperature for 60 min and Detection Antibody for 30 min. After adding Developer Solution and Stop Solution, the color of sample RNA changed into yellow and the absorbance was read at a wavelength of 450 nm (OD450). The relative RNA m6A methylation level was calculated based on the standard curve.

### 4.8. Immunofluorescence

BMSCs were seeded on glass coverslips. At 3 days post the induction of osteogenic differentiation, the coverslips were washed and fixed, followed by permeabilization, blocking, and incubating with primary anti-m6A antibody (A19841, ABclonal, Wuhan, China) at 4 °C overnight. On the second day, the coverslips were washed and incubated with 594 conjugated secondary antibodies (E031220, EarthOx, San Francisco, CA, USA). In addition, DAPI (F6057, Sigma-Aldrich, St. Loius, MO, USA) was used to identify the nuclei of BMSCs and images were captured with a laser scanning confocal microscope (FV3000, Olympus, Tokyo, Japan).

### 4.9. SAM-ELISA

The levels of SAM in BMSCs were measured by the SAM ELISA Kit (MET-5152, Cell Biolabs, San Diego, CA, USA) according to the manufacturer’s instructions. Briefly, 50 μL cell lysate was added to the SAM Conjugate-coated plate and incubated at room temperature for 10 min. A total of 50 μL diluted anti-SAM antibody was then added and incubated at room temperature for 60 min. The cell lysate and diluted anti-SAM antibody were removed, following incubation with diluted secondary antibody HRP Conjugate for 60 min and Substrate Solution for 30 min. After adding Stop Solution, the absorbance was read immediately at a wavelength of 450 nm (OD450). The SAM level was calculated based on the standard curve.

### 4.10. Construction of Plasmids and Transfection of Virus

METTL3 short hairpin RNA (METTL3sh) plasmid was constructed by Genechem Company (Shanghai, China). METTL3sh, 5′-GCTGCACTTCAGACGAATTAT-3′ plasmids were subcloned into LV3 lentiviral vector. Virus transfection was conducted as previously described [33].

### 4.11. Statistics Analysis

Results are presented as mean ± SEM Statistical analysis was performed using SPSS, version 19.0. Differences between two groups were examined by unpaired 2-tailed Student’s *t*-test. Differences among more than two groups were examined by one-way ANOVA test, followed by Tukey’s multiple comparisons test. In all cases, *p* < 0.05 was considered as statistically significant.

## Figures and Tables

**Figure 1 ijms-26-05233-f001:**
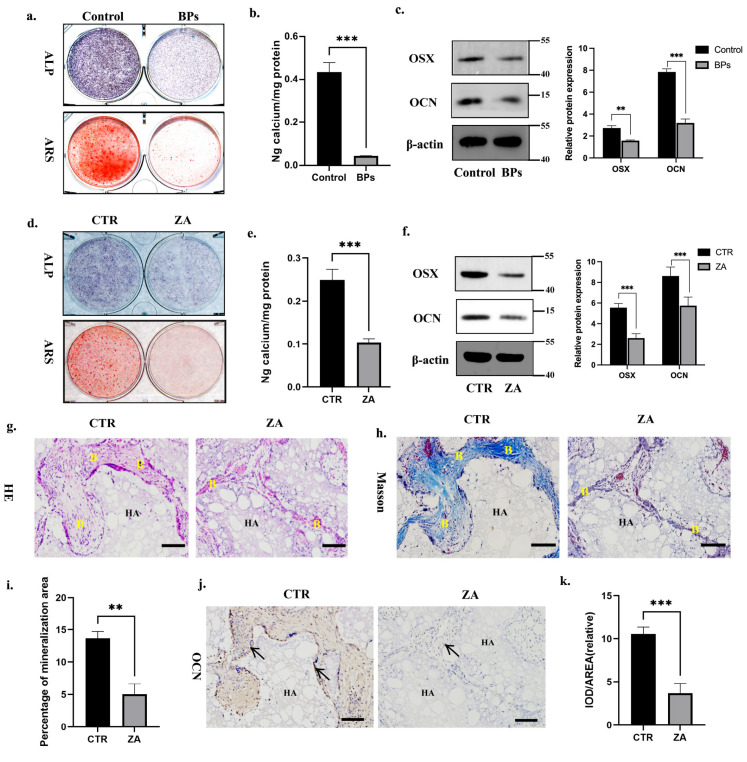
BPs impaired orofacial BMSCs’ osteogenic differentiation ability. (**a**). Orofacial BMSCs were extracted from rats’ mandible bones following the rats receiving intraperitoneal injection of ZA or saline. ALP and ARS staining were performed on days 7 and 14 of osteogenic differentiation, respectively. (**b**). Calcium ion quantification of ARS staining. (**c**). Immunoblotting images of OSX/OCN on day 7. (**d**). Human orofacial BMSCs were isolated from donors’ mandible bones and pretreated with or without 5 µM ZA for 3 days. ALP and ARS staining were performed on days 7 and 14 of osteogenic differentiation, respectively. (**e**). Calcium ion quantification of ARS staining. (**f**). Immunoblotting images of OSX/OCN on day 7. (**g**). Human orofacial BMSCs were pretreated with or without 5 µM ZA for 3 days and then transplanted subcutaneously into the back of nude mice, HE staining was performed 8 weeks post-transplantation. (**h**). Masson staining was performed 8 weeks post-transplantation. (**i**). The quantitative analyses of the mineralization area in HE staining images. (**j**). Immunohistochemical staining of OCN was performed 8 weeks post-transplantation. (**k**). The quantitative analyses of the OCN staining. Scale bar: 100 μm. B: bone-like tissue; black arrow: OCN-positive BMSCs; HA: hydroxyapatite tricalcium carrier. N = 5 nude mice per group. Data are expressed as mean ± SEM; ** *p* < 0.01, *** *p* < 0.001, CTR/Control vs. ZA/BPs.

**Figure 2 ijms-26-05233-f002:**
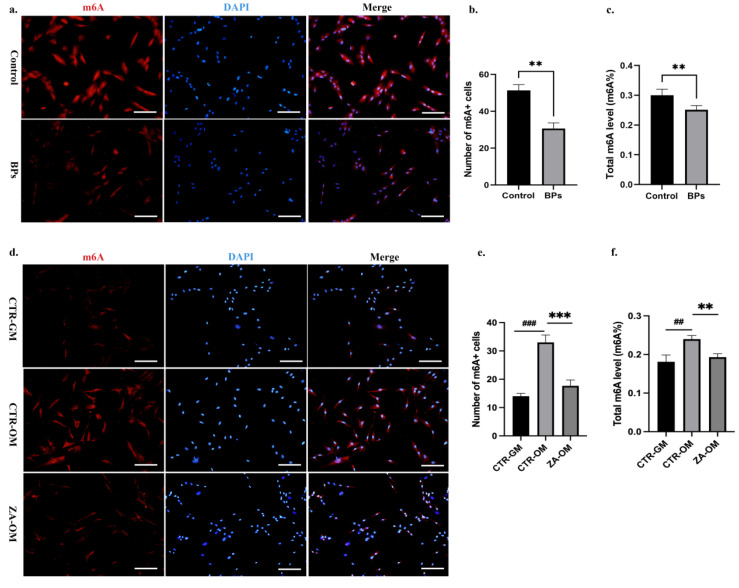
BPs decreased RNA m6A methylation level in BMSCs. (**a**). BMSCs were extracted from saline or ZA-injected rats and cultured in osteogenic induction medium. On day 3, the nucleus and m6A were labelled with DAPI and anti-m6A antibodies, respectively. (**b**). The quantitative analyses of m6A-positive cells. (**c**). The quantification of global m6A methylation level in BMSCs was detected by conducting a colorimetric m6A quantification assay on day 7. (**d**). Human orofacial BMSCs were cultured with growth medium (GM) or osteogenic induction medium (OM) after being pretreated with or without 5 µM ZA for 3 days. On day 3, the nucleus and m6A were labelled with DAPI and anti-m6A antibodies, respectively. (**e**). The quantitative analyses of m6A-positive cells. (**f**). The quantification of global m6A methylation level in BMSCs was detected by conducting a colorimetric m6A quantification assay on day 7. Scale bars: 200 μm. Data are expressed as mean ± SEM; ** *p* < 0.01, *** *p* < 0.001, CTR-OM/Control vs. ZA-OM/BPs. ## *p* < 0.01, ### *p* < 0.001, CTR-GM vs. CTR-OM.

**Figure 3 ijms-26-05233-f003:**
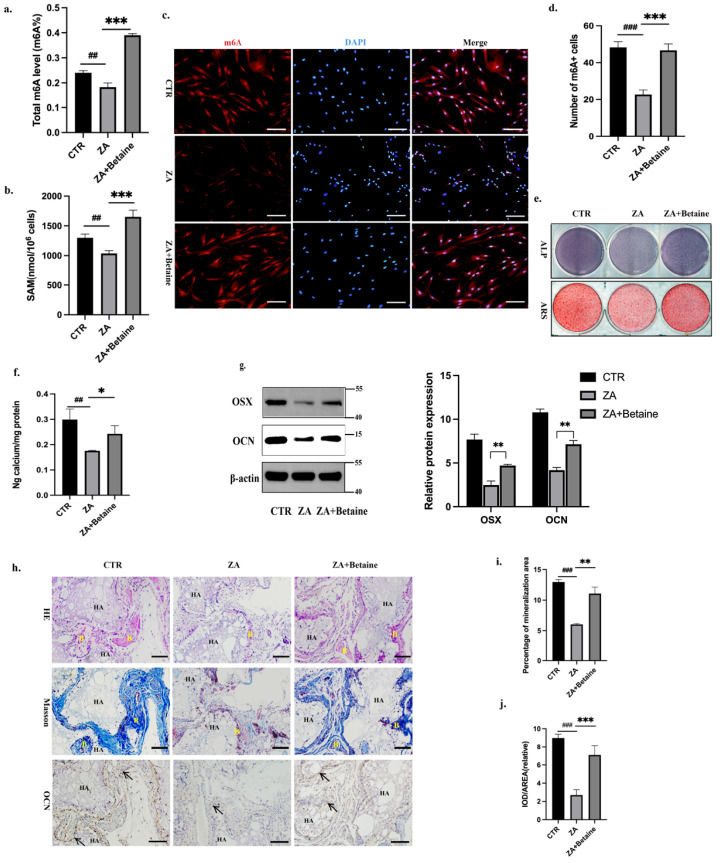
Betaine rescued BP-induced BMSC dysfunction in vitro. (**a**). Human orofacial BMSCs were pretreated with or without ZA for 3 days and then cultured in osteogenic induction medium with or without 10 mM betaine. Colorimetric quantification assay was performed on day 7 to detect the global m6A methylation level in BMSCs. (**b**). Colorimetric quantification assay was performed on day 7 to detect the SAM level in BMSCs. (**c**). On day 3, the nucleus and m6A were labelled with DAPI and anti-m6A antibodies, respectively. (**d**). The quantitative analyses of m6A-positive cells. (**e**). ALP and ARS staining were performed on day 7 and 14 of osteogenic differentiation, respectively. (**f**). Calcium ion quantification of ARS staining. (**g**). Immunoblotting images of OSX/OCN on day 7. (**h**). Human orofacial BMSCs were pretreated with or without ZA and betaine for 3 days and then transplanted subcutaneously into the backs of nude mice. HE/Masson staining and immunohistochemical staining of OCN were performed 8 weeks post-transplantation. (**i**). The quantitative analyses of the mineralization area. (**j**). The quantitative analyses of the OCN staining. Scale bar: 100 μm. B: bone-like tissue; black arrow: OCN-positive BMSCs; HA: hydroxyapatite tricalcium carrier. N = 5 nude mice per group. Data are expressed as mean ± SEM; * *p* < 0.05, ** *p* < 0.01, *** *p* < 0.001, ZA vs. ZA+Betaine. ## *p* < 0.01, ### *p* < 0.001, CTR vs. ZA.

**Figure 4 ijms-26-05233-f004:**
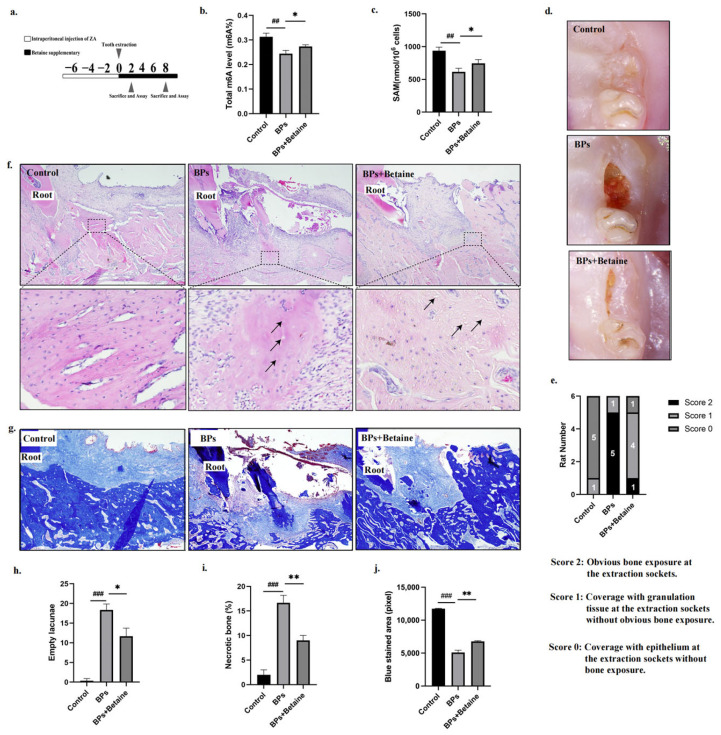
Betaine alleviated osteonecrosis of the jaw in BRONJ rat model at 2 weeks post-tooth extraction. (**a**). Schematic overview of study procedures. (**b**). BMSCs were extracted from rats’ mandible bones. Colorimetric quantification assay was performed to detect the global m6A methylation level in BMSCs. (**c**). Colorimetric quantification assay was performed to detect the SAM level. (**d**). Representative images of mucosal healing at the extraction sites. (**e**). The score of mucosal healing. (**f**,**g**). Representative HE-/Masson-stained images of tooth extraction sockets. (**h**). The number of osteocyte lacunae without nuclear staining were quantified as empty lacunae. The quantification of empty lacunae was performed. (**i**). Areas containing 10 empty lacunae were quantified as necrotic bone and were presented as a percentage of the total bone present. (**j**). The quantitative analyses of blue-stained area. Black arrow: empty lacunae; N = 6 rats per group. Data are expressed as mean ± SEM; * *p* < 0.05, ** *p* < 0.01, BPs vs. BPs+Betaine. ## *p* < 0.01, ### *p* < 0.001, Control vs. BPs.

**Figure 5 ijms-26-05233-f005:**
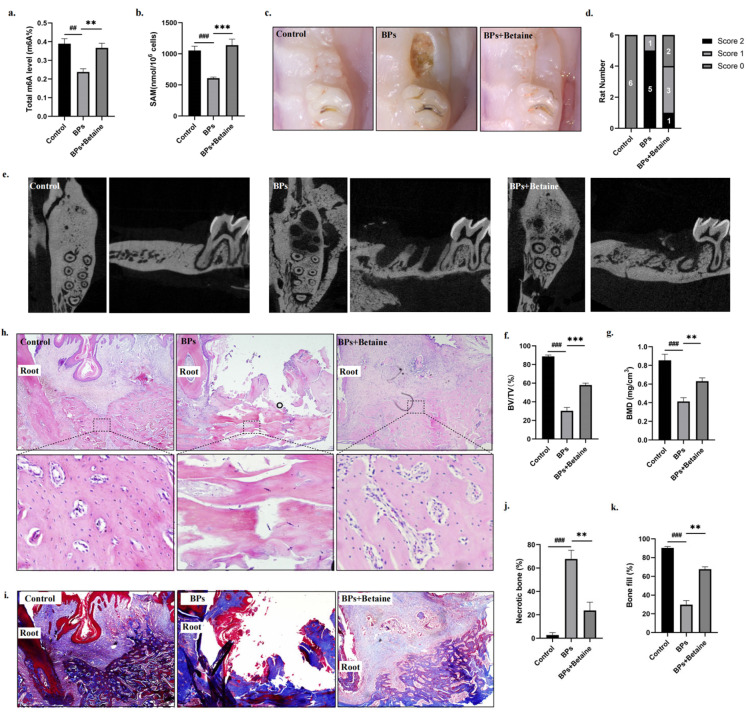
Betaine alleviated osteonecrosis of the jaw in BRONJ rat model at 8 weeks post-tooth extraction. (**a**). BMSCs were extracted from rats’ mandible bones. Colorimetric quantification assay was performed to detect the global m6A methylation level in BMSCs. (**b**). Colorimetric quantification assay was performed to detect the SAM level. (**c**). Representative images of mucosal healing at the extraction sites. (**d**). The score of mucosal healing. (**e**). Representative micro-CT images. (**f**,**g**). The relative bone volume (BV/TV) and bone mineral density (BMD) were measured. (**h**,**i**). Representative HE-/Masson-stained images of tooth extraction sockets. (**j**). Areas containing 10 empty lacunae were quantified as necrotic bone and were presented as a percentage of the total bone present. (**k**). The quantitative analyses of blue-stained area. N = 6 rats per group. Data are expressed as mean ± SEM; ** *p* < 0.01, *** *p* < 0.001, BPs vs. BPs+Betaine. ## *p* < 0.01, ### *p* < 0.001, Control vs. BPs.

**Figure 6 ijms-26-05233-f006:**
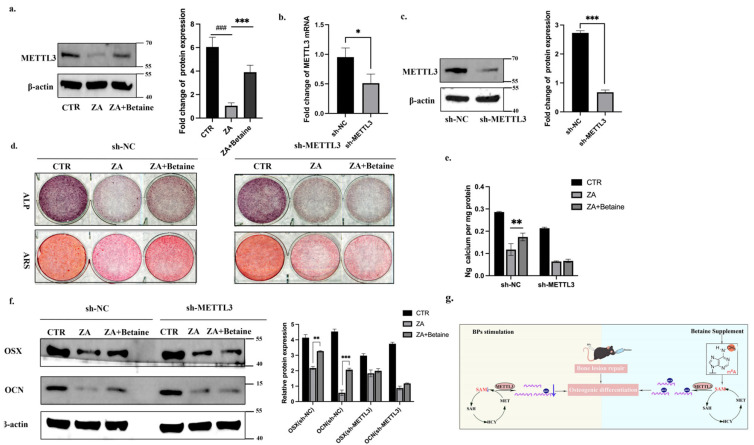
Betaine regulated BMSCs function in an m6A-METTL3-dependent manner. (**a**). Human orofacial BMSCs were pretreated with or without ZA for 3 days and then cultured in osteogenic induction medium with or without 10 mM betaine. Immunoblotting of METTL3 was performed on day 7 of osteogenic differentiation. (**b**,**c**). The mRNA and protein expression level of METTL3 from sh-NC/sh-METTL3 Group. (**d**). sh-NC/sh-METTL3 BMSCs were pretreated with or without ZA for 3 days and then cultured in osteogenic induction medium with or without 10 mM betaine. ALP and ARS staining were performed on day 7 and 14 of osteogenic differentiation, respectively. (**e**). Calcium ion quantification of ARS staining. (**f**). Representative immunoblotting images of OSX/OCN on day 7. (**g**). Molecular mechanism diagram. Data are expressed as mean ± SEM; * *p* < 0.05, ** *p* < 0.01, *** *p* < 0.001, sh-NC/ZA vs. sh-METTL3/ZA+Betaine. ### *p* < 0.001, CTR vs. ZA.

## Data Availability

All data used to support the findings of this study are included within the article.

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
