# Peer review of "Betaine Alleviates Bisphosphonate-Related Osteonecrosis of the Jaw by Rescuing BMSCs Function in an m6A-METTL3-Dependent Manner"

_ijms, 2025, doi:10.3390/ijms26115233_

Round 1
Reviewer 1 Report
Comments and Suggestions for Authors
1, How the Betaine dose was determined, the authors should add the results after treatment with different concentrations in in vitro experiments.
2, The figure should be below the corresponding description text.
3, The data in the figure notes are expressed as mean±SEM, while in 4.11 it is mean±SD, the authors should verify this information.
4, The animal experiments do not indicate the statistical number of animals.
5, Methylation of DNA affects the transcription and expression of genes by affecting the transcription and expression of genes, the authors need to add the qPCR results of genes such as OSX OCN METTL3.
6, The authors should add a molecular mechanism diagram.
7, The band for OSX in Figure 6f is incomplete and should be replaced.
8, All WB experimental results should be quantified.
9, In the discussion section, the authors should compare the advantages of Betaine with the reported mitigants.
10, The Methods section is very simple and the reader sees it and does not replicate it, e.g., 4.1 does not state the composition and ratio of the cell culture medium. the solvents configured with Zoledronic acid and Betaine, and whether the solvents were verified to be non-toxic to the cells.
Author Response
Comments 1: How the Betaine dose was determined, the authors should add the results after treatment with different concentrations in in vitro experiments. |
Response 1: Thank you very much for your kind suggestion! We determined Betaine dose according to the previously published articles. 10mM Betaine has been proved to promote osteogenic differentiation in dental pulp stem cells [1], adipose-derived stem cells [2], mesenchymal stem cells [3] and osteoblasts derived from trabecular bone [4], therefore we supplemented orofacial BMSCs with 10mM Betaine in this study. [1] Kornsuthisopon C, Nantanapiboon D, Rochanavibhata S, Nowwarote N, Namangkalakul W, Osathanon T. Betaine promotes osteogenic differentiation in immortalized human dental pulp-derived cells. BDJ Open. 2022 Oct 7;8(1):31. doi: 10.1038/s41405-022-00123-7. [2] Tabatabai TS, Haji-Ghasem-Kashani M, Nasiri M. In vitro osteogenic induction of human adipose stem cells co-treated with betaine/osteogenesis differentiation medium. Mol Biol Res Commun. 2021;10:93–103. doi: 10.22099/mbrc.2021.39354.1578. [3] Jing Y, Zhou J, Guo F, Yu L, Ren X, Yin X. Betaine regulates adipogenic and osteogenic differentiation of hAD-MSCs. Mol Biol Rep. 2023 Jun;50(6):5081-5089. doi: 10.1007/s11033-023-08404-6. Epub 2023 Apr 26. Erratum in: Mol Biol Rep. 2023 Nov;50(11):9743. doi: 10.1007/s11033-023-08779-6. [4] Villa I, Senesi P, Montesano A, Ferraretto A, Vacante F, Spinello A, et al. Betaine promotes cell differentiation of human osteoblasts in primary culture. J Transl Med. 2017;15:132. doi: 10.1186/s12967-017-1233-5. |
Comments 2: The figure should be below the corresponding description text. |
Response 2: Thanks for the suggestion! We have placed the figures below the corresponding description text in the revised manuscript. |
Comments 3: The data in the figure notes are expressed as mean±SEM, while in 4.11 it is mean±SD, the authors should verify this information. |
Response 3: We are very grateful for your thorough reviews and we apologize for the errors in 4.11. We have corrected 4.11. Statistics Analysis part in the revised manuscript. |
Comments 4: The animal experiments do not indicate the statistical number of animals. |
Response 4: We are very grateful for your thorough reviews and we have incorporated the statistical number of animals in Corresponding Figure Legends and 4.2 Animals and Treatments part. |
Comments 5: Methylation of DNA affects the transcription and expression of genes by affecting the transcription and expression of genes, the authors need to add the qPCR results of genes such as OSX OCN METTL3. |
Response 5: Thanks for the suggestion! We are totally agree with your comments that methylation modification affects gene transcription, however, in this study we did not figure out the downstream genes which were betaine-methylated. In one of our previous study, we applied m6A epitranscriptomic microarray to demonstrate CORIN is one of the downstream methylation-changed genes under BPs stimulation [1]. In a study from other group, they proved HIF-1α was one of downstream genes which were betaine-methylated in BMSCs [2]. More work need to be done to figure out the betaine-methylated genes in orofacial BMSCs and We have revised the Discussion part in the manuscript. [1] Jin Y, Han X, Wang Y, Fan Z. METTL7A-mediated m6A modification of corin reverses bisphosphonates-impaired osteogenic differentiation of orofacial BMSCs. Int J Oral Sci. 2024 May 23;16(1):42. doi: 10.1038/s41368-024-00303-1. [2] Zhang W, Bai Y, Hao L, Zhao Y, Zhang L, Ding W, Qi Y, Xu Q. One-carbon metabolism supports S-adenosylmethionine and m6A methylation to control the osteogenesis of bone marrow stem cells and bone formation. J Bone Miner Res. 2024 Sep 2;39(9):1356-1370. doi: 10.1093/jbmr/zjae121. |
Comments 6: The authors should add a molecular mechanism diagram. |
Response 6: Thank you for your constructive suggestions! We have incorporated a molecular mechanism diagram in Figure 6 which has been highlighted. |
Comments 7: The band for OSX in Figure 6f is incomplete and should be replaced. |
Response 7: Thank you for your suggestion! Since we consider that the band for protein is complete and have little influence on quantification, we do not replace this band in the revised manuscript. |
Comments 8: All WB experimental results should be quantified. |
Response 8: Thank you for your constructive suggestions! We have quantified all WB experimental results and updated all Figures in the revised manuscript. |
Comments 9: In the discussion section, the authors should compare the advantages of Betaine with the reported mitigants. |
Response 9: Thank you for your suggestions and we are totally agree with your comments. We have compared the advantages of betaine with reported preventive or treatment strategies in the Discussion part. |
Comments 10: The Methods section is very simple and the reader sees it and does not replicate it, e.g., 4.1 does not state the composition and ratio of the cell culture medium. the solvents configured with Zoledronic acid and Betaine, and whether the solvents were verified to be non-toxic to the cells. |
Response 10: Thank you for your constructive comments and we are totally agree with your suggestions! We have incorporated more detailed information in Methods part. |

Reviewer 2 Report
Comments and Suggestions for Authors
General comments:
This paper describes a study on the efficacy and mechanisms of betaine for treating bisphosphonate-related osteonecrosis (BRONJ). The strong efficacy of betaine suggests its potential as a therapeutic agent for BRONJ. However, several concerns are outlined below.
Detailed comments:
- Throughout the manuscript, the terms “Control” and ‘CTR’ are used interchangeably in the figures. Is there a difference between the two terms?
- The term “BPs” is also used in the figures throughout the manuscript. Does this refer to ZA?
- Betaine is only used in combination with ZA. Please accurately evaluate the effects of betaine alone and present the data.
Author Response
Comments 1: Throughout the manuscript, the terms “Control” and ‘CTR’ are used interchangeably in the figures. Is there a difference between the two terms? |
Response 1: In this study, we extracted orofacial BMSCs both from rats and human. The term “Control” referred to BMSCs extracted from rats’ mandible bone which have received intraperitoneal injection of saline (0.1mg/kg) for 6 weeks (3 times/week). The term “CTR” referred to BMSCs extracted from healthy human donors’ mandible bone, which were not pretreated with ZA 3 days before the induction of osteogenic differentiation. |
Comments 2: The term “BPs” is also used in the figures throughout the manuscript. Does this refer to ZA? |
Response 2: Yes. BPs are divided into nitrogen and non-nitrogen BPs. ZA is a nitrogen-containing third-generation BPs, which is most widely used in the clinic treatment. We use terms “BPs” and “ZA” to distinguish two sources of BMSCs. The term “BPs” referred to BMSCs extracted from rats’ mandible bone which have received intraperitoneal injection of ZA (0.1mg/kg) for 6 weeks (3 times/week). The term “ZA” referred to BMSCs extracted from healthy human donors’ mandible bone, which were pretreated with ZA 3 days before the induction of osteogenic differentiation. |
Comments 3: Betaine is only used in combination with ZA. Please accurately evaluate the effects of betaine alone and present the data. |
Response 3: Thank you for your constructive suggestions! The promotion effect of betaine on osteogenic differentiation has been demonstrated in various cell lines, including dental pulp stem cells [1], adipose-derived stem cells [2], mesenchymal stem cells [3] and osteoblasts derived from trabecular bone [4]. We have incorporated the effects of betaine alone in the Supplementary Materials. [1] Kornsuthisopon C, Nantanapiboon D, Rochanavibhata S, Nowwarote N, Namangkalakul W, Osathanon T. Betaine promotes osteogenic differentiation in immortalized human dental pulp-derived cells. BDJ Open. 2022 Oct 7;8(1):31. doi: 10.1038/s41405-022-00123-7. [2] Tabatabai TS, Haji-Ghasem-Kashani M, Nasiri M. In vitro osteogenic induction of human adipose stem cells co-treated with betaine/osteogenesis differentiation medium. Mol Biol Res Commun. 2021;10:93–103. doi: 10.22099/mbrc.2021.39354.1578. [3] Jing Y, Zhou J, Guo F, Yu L, Ren X, Yin X. Betaine regulates adipogenic and osteogenic differentiation of hAD-MSCs. Mol Biol Rep. 2023 Jun;50(6):5081-5089. doi: 10.1007/s11033-023-08404-6. Epub 2023 Apr 26. Erratum in: Mol Biol Rep. 2023 Nov;50(11):9743. doi: 10.1007/s11033-023-08779-6. [4] Villa I, Senesi P, Montesano A, Ferraretto A, Vacante F, Spinello A, et al. Betaine promotes cell differentiation of human osteoblasts in primary culture. J Transl Med. 2017;15:132. doi: 10.1186/s12967-017-1233-5. |

Reviewer 3 Report
Comments and Suggestions for Authors
RE: ijms-3620730
Betaine alleviates bisphosphonate-rerated osteonecrosis of the jaw by rescuing BMSCs function in an m6A-METTL3-dependent manner
There are some reports that bisphosphonate (BP) could induce osteonecrosis of the jaw (MRONJ) by impairing function of the bone marrow stromal cells (BMSC). This study verified that betaine could suppresses BP-impaired osteodifferentiation of BMSCs, and these results may contribute to the understanding of the pathogenesis of MRONJ and to the development of therapeutic methods. It was considered that the study was basically logical. Only some minor questions were listed below.
Figure 1-g, h. Although the footnote showed the star meant the bone like tissue, there were not clear. Please replace them to the clear ones.
Figures 1 and 2, Why cells were used on 3 days culture in figure 2 and on 7 and 14 days in figure 1.
How was the amount of the betaine decided in this study?
Author Response
Comments 1: Figure 1-g, h. Although the footnote showed the star meant the bone like tissue, there were not clear. Please replace them to the clear ones. |
Response 1: Thank you for your constructive suggestions! We have replaced with more clear indicators in the revised manuscript. |
Comments 2: Figures 1 and 2, Why cells were used on 3 days culture in figure 2 and on 7 and 14 days in figure 1. |
Response 2: We have observed that the altered expression of bone formation-related genes tend to be stable on the seventh day during osteogenic differentiation course, therefore we performed m6A-ELISA and Westernblot on day 7. ALP staining is an early osteogenic indicator which is generally performed on day 7, while ARS staining is a late osteogenic indicator which is usually performed on day 14. During osteogenic differentiation course, BMSCs witnessed cell elongation and increased cell density, which might have a influence on calculating m6A-positive cells. Therefore, we performed m6A-immunofluorescence on day 3 due to proper cell density and cell morphology, which is consistent with Zhang’s study [1]. [1] Zhang W, Bai Y, Hao L, Zhao Y, Zhang L, Ding W, Qi Y, Xu Q. One-carbon metabolism supports S-adenosylmethionine and m6A methylation to control the osteogenesis of bone marrow stem cells and bone formation. J Bone Miner Res. 2024 Sep 2;39(9):1356-1370. doi: 10.1093/jbmr/zjae121. |
Comments 3: How was the amount of the betaine decided in this study? |
Response 3: The Betaine dose was determined according to the previously published articles. 10mM Betaine has been proved to promote osteogenic differentiation in dental pulp stem cells [1], adipose-derived stem cells [2], mesenchymal stem cells [3] and osteoblasts derived from trabecular bone [4], therefore we supplemented orofacial BMSCs with 10mM Betaine in this study. [1] Kornsuthisopon C, Nantanapiboon D, Rochanavibhata S, Nowwarote N, Namangkalakul W, Osathanon T. Betaine promotes osteogenic differentiation in immortalized human dental pulp-derived cells. BDJ Open. 2022 Oct 7;8(1):31. doi: 10.1038/s41405-022-00123-7. [2] Tabatabai TS, Haji-Ghasem-Kashani M, Nasiri M. In vitro osteogenic induction of human adipose stem cells co-treated with betaine/osteogenesis differentiation medium. Mol Biol Res Commun. 2021;10:93–103. doi: 10.22099/mbrc.2021.39354.1578. [3] Jing Y, Zhou J, Guo F, Yu L, Ren X, Yin X. Betaine regulates adipogenic and osteogenic differentiation of hAD-MSCs. Mol Biol Rep. 2023 Jun;50(6):5081-5089. doi: 10.1007/s11033-023-08404-6. Epub 2023 Apr 26. Erratum in: Mol Biol Rep. 2023 Nov;50(11):9743. doi: 10.1007/s11033-023-08779-6. [4] Villa I, Senesi P, Montesano A, Ferraretto A, Vacante F, Spinello A, et al. Betaine promotes cell differentiation of human osteoblasts in primary culture. J Transl Med. 2017;15:132. doi: 10.1186/s12967-017-1233-5. |

Round 2
Reviewer 1 Report
Comments and Suggestions for Authors
The authors have completed revisions based on the comments. The quality of this thesis has been significantly improved.
Reviewer 2 Report
Comments and Suggestions for Authors
The authors appropriately addressed most of the points raised in my previous review. Therefore, there are no longer any concerns about publishing the article in the journal.